# Fully inkjet-printed Ag₂Se flexible thermoelectric devices for sustainable power generation

Yan Liu[1,10], Qihao Zhang [2,10] ✉, Aibin Huang[3,4,10], Keyi Zhang[1], Shun Wan[5], Hongyi Chen[6], Yuntian Fu[1], Wusheng Zuo[1], Yongzhe Wang[3,4], Xun Cao [3,4] ✉, Lianjun Wang [1,7] ✉, Uli Lemmer[2,8] & Wan Jiang [1,9] ✉

Flexible thermoelectric devices show great promise as sustainable power units for the exponentially increasing self-powered wearable electronics and ultra-widely distributed wireless sensor networks. While exciting proof-of-concept demonstrations have been reported, their large-scale implementation is impeded by unsatisfactory device performance and costly device fabrication techniques. Here, we develop Ag₂Se-based thermoelectric films and flexible devices via inkjet printing. Large-area patterned arrays with microscale resolution are obtained in a dimensionally controlled manner by manipulating ink formulations and tuning printing parameters. Printed Ag₂Se-based films exhibit (00 $l$)-textured feature, and an exceptional power factor (1097 $\mu Wm^{-1}K^{-2}$ at 377 K) is obtained by engineering the film composition and microstructure. Benefiting from high-resolution device integration, fully inkjet-printed Ag₂Se-based flexible devices achieve a record-high normalized power (2 $\mu WK^{-2}cm^{-2}$) and superior flexibility. Diverse application scenarios are offered by inkjet-printed devices, such as continuous power generation by harvesting thermal energy from the environment or human bodies. Our strategy demonstrates the potential to revolutionize the design and manufacture of multi-scale and complex flexible thermoelectric devices while reducing costs, enabling them to be integrated into emerging electronic systems as sustainable power sources.

Flexible thermoelectric materials and devices have attracted extensive attention due to their ability to generate electricity directly by harvesting thermal energy from the environment or human bodies[1–3]. With the advantages of small size, lightweight, no moving parts, high reliability, and conformal contact with arbitrarily shaped heat sources, they show great promise as sustainable power supply units for the exponentially increasing number of smart devices such as self-powered portable/wearable low-energy electronics[4], ultra-widely

¹State Key Laboratory for Modification of Chemical Fibers and Polymer Materials, College of Materials Science and Engineering, Donghua University, 201620 Shanghai, China. ²Light Technology Institute, Karlsruhe Institute of Technology, Engesserstrasse 13, 76131 Karlsruhe, Germany. ³State Key Laboratory of High Performance Ceramics and Superfine Microstructure, Shanghai Institute of Ceramics, Chinese Academy of Sciences, 200050 Shanghai, China. ⁴Center of Materials Science and Optoelectronics Engineering, University of Chinese Academy of Sciences, 100049 Beijing, China. ⁵Center for High Pressure Science and Technology Advanced Research (HPSTAR), 201203 Shanghai, China. ⁶College of Chemistry and Chemical Engineering, Central South University, 410083 Changsha, China. ⁷Engineering Research Center of Advanced Glasses Manufacturing Technology, Ministry of Education, Donghua University, 201620 Shanghai, China. ⁸Institute of Microstructure Technology (IMT), Karlsruhe Institute of Technology (KIT), Hermann-von-Helmholtz-Platz 1, 76344 Eggenstein-Leopoldshafen, Germany. ⁹Institute of Functional Materials, Donghua University, 201620 Shanghai, China. ¹⁰These authors contributed equally: Yan Liu, Qihao Zhang, Aibin Huang. ✉e-mail: qihao.zhang@kit.edu; cxun@mail.sic.ac.cn; wanglj@dhu.edu.cn; wanjiang@dhu.edu.cn

distributed wireless sensor network nodes and Internet of things (IoT) devices[5]. Generally, there are two ways to fabricate flexible thermoelectric devices: connecting rigid bulk thermoelectric pellets by flexible electrodes and substrates[6–8]; and developing flexible thermoelectric materials such as conducting polymers[9–11], inorganic films[12–14], organic-inorganic composites[1,15,16], and ductile semiconductors[2,17]. The devices fabricated from films offer superior structural deformation owing to their inherent flexibility, but they usually produce lower output power densities, mainly due to the limited thickness of the films and the poor thermoelectric properties.

In addition to performance metrics, another crucial step towards the commercialization and industrialization of flexible thermoelectric technology in areas such as electronics and energy harvesting is to achieve reliable large-scale production and processability of these films and devices. Traditionally, film-based flexible thermoelectric devices have been manufactured primarily through photolithography combined with physical or chemical deposition processes[5,18,19]. However, these methods suffer from disadvantages including multi-step procedures, expensive equipment, and the production of large amounts of environmentally harmful waste. Other methods such as vacuum filtration[12] and spin/spray coating[9] have also been used for film-based device fabrication, but they offer poor control of the thickness and roughness of the films and have limited design flexibility and low levels of integration. Moreover, the films are prepared independently of the devices. As a result, performance is greatly reduced when the films are assembled into the devices. As alternatives, ink-based printing techniques, including screen printing, dispenser printing, aerosol jet printing, roll-to-roll printing, and inkjet printing are gaining prominence to facilitate efficient, versatile, and scalable manufacturing[20–22]. Among them, inkjet printing is particularly advantageous in the manufacture of film devices owing to its low cost, easily changeable digital printing patterns, and low material consumption[22]. It enables the accurate deposition of micro- and nano-materials into functional arrangements in a non-impact, additive patterning and maskless approach, becoming a forefront technique for advanced miniaturized electronics with customized patterns and high precision. Over the past decade, inkjet printing of flexible thermoelectric materials including PEDOT[23,24], Bi-Te alloys[25,26], graphene[27], and TiS₂(HA)ₓ[28] have been reported, demonstrating a versatile platform to transform material building blocks into functional devices. However, power generation performance of the corresponding inkjet-printed thermoelectric devices is not competitive with that of conventionally fabricated devices[5,21,22]. In addition, inkjet printing of structures at the micrometer scale remains challenging due to the limited availability of printable and high-performance thermoelectric inks. Innovative ink formulations that not only offer excellent processability and stability, but also enable on-demand manipulation of electron and phonon transport properties are essential for the development of next-generation, high-performance inkjet-printed thermoelectric devices

Ag₂Se exhibits excellent thermoelectric properties below 100 °C, making it a potential candidate for generating electricity from low-grade heat[12]. A number of groups are currently conducting research to prepare Ag₂Se-based flexible devices, and this has led to significant progress[12,29–32]. However, the degree of device integration, power-generation performance, and flexibility are still far from expectations[33]. For example, some Ag₂Se films show high power factors, but they exhibit poor control of the thickness and roughness of the films and have limited design flexibility and low levels of device integration. In addition, despite exceptional power factors presented by the films, the power density of corresponding devices is quite low. This is mainly due to the fact that these films are prepared independently of the devices, resulting in a significant loss of performance during the assembly of the films into the devices.

To address the above issues, we thereby focus on inkjet printing technology and take Ag₂Se materials as research objects. First of all, we develop additive-free Ag₂Se-based inks. By manipulating Ag₂Se-based ink formulations and optimizing printing parameters, we obtain pattern arrays with high resolution in a controlled manner. For example, multiple parallel lines with a line width of 150 μm and a gap of 100 μm are accurately achieved. Benefiting from excellent ink printability and composition design, the printed films exhibit a high power factor of 1097 μW m⁻¹ K⁻² at 377 K, more than five times that of state-of-the-art inkjet-printed materials. To exploit the capabilities of the inkjet printing process, a number of flexible thermoelectric devices with leg widths and lengths ranging from micrometers to millimeters are fabricated. The filling factor of our inkjet-printed devices can reach up to 81%, and the density of leg integration gets as high as 125 legs per square centimeter. This unique patterning capability and high-resolution device integration has rarely been reported for flexible thermoelectric devices because it is quite challenging to achieve via commonly used film preparation methods such as spin coating, sputtering, thermal evaporation, and screen printing. As a result, our Ag₂Se-based inkjet-printed devices demonstrate a record-high normalized power ($2\,\mu W K^{-2}\,cm^{-2}$) and exceptional device flexibility (surviving 3,000 bending tests at bending radii of 3–4 mm), proving great potential for applications in powering portable/wearable and low-power electronics.

## Results and discussion
### Ink formulation and printability
Ag₂Se nanoparticles were synthesized by a solvothermal reaction, as detailed in Methods. Powder X-ray diffraction (XRD) patterns (Supplementary Fig. 1a) show that the diffraction peaks of as-synthesized Ag₂Se nanoparticles can be indexed to β-Ag₂Se phase (PDF #24-1041) with the space group of $P2_12_12_1$. Scanning electron microscopy (SEM), transmission electron microscopy (TEM), and atomic force microscopy (AFM) images indicate that Ag₂Se nanoparticles are pellet-like in shape, featuring a narrow size distribution and an average particle lateral diameter of 108 nm with an average thickness of 50 nm (Fig. 1a and Supplementary Fig. 1). Representative elemental mapping results (Supplementary Fig. 1d) further reveal a homogeneous distribution of the constituent elements within Ag₂Se nanoparticles. High-resolution TEM image shows lattice spacings of 3.08 Å, 3.66 Å, and 2.30 Å, corresponding to the (102), (020), and (122) lattice planes of orthorhombic Ag₂Se, as confirmed by fast Fourier transform (FFT) pattern (Supplementary Fig. 1e). These phase and microstructure characterizations confirm the synthesis of nanoscale orthorhombic Ag₂Se with particle dimensions that fully meets the requirements for drop-on-demand inkjet (solute size less than 1/10th of the nozzle diameter (21 μm in this work) to avoid clogging or blocking during printing[34]).

Then, Ag₂Se ink was prepared by dispensing Ag₂Se nanoparticles in anhydrous ethanol, followed by sonication for 2 h (see "Methods"). The resulting ink exhibits good dispersion and stability, as there is no obvious sedimentation or aggregation in the resting experiments (Fig. 1b). Such good stability is due to the presence of poly-vinylpyrrolidone (PVP) attached to the surface of Ag₂Se nanoparticles (Supplementary Fig. 1e), which acts as an effective steric stabilizer to stabilize the dispersion system. In addition, the ink is free of toxic additives, environmentally friendly, and sustainable, making it suitable for industrial-scale production. To be inkjet printable, the ink must have specific physical properties, such as viscosity, surface tension, and density, which are within suitable ranges for a fixed nozzle diameter. In general, the inverse Ohnesorge number $Z$ can be used to determine the jettability of the inks (Supplementary Note 1). It is proposed that the $Z$ has to be in the range of 1–14 for stable drop generation[22]. For a fixed nozzle diameter of 21 μm (DMC 11610 cartridge) in this work, our Ag₂Se ink with a concentration of 10 mg/mL gives a $Z$ value of 13.5 (Supplementary Table 1). By applying a single-peak waveform with a jetting frequency of up to 2.5 kHz and a maximum voltage of 18 V (Fig. 1c and Supplementary Fig. 2), the drop

velocity is determined to be 4.5 m·s⁻¹. The corresponding *Re* and *We* are both within the printable region constructed in the *Re-We* parameter space[35,36] (Fig. 1d), ensuring stable droplet formation without any satellite tails or nozzle blockage (Fig. 1e). In addition, to get rid of the coffee effect, a substrate temperature of 40 °C and a nozzle temperature of 30 °C are used in the printing process.

The size of deposited droplets, which is an important parameter that affects the continuity and integrity of the printed layer, is then investigated by jetting individual drops on polyimide and photographic paper substrates. As shown in Fig. 1f and Supplementary Fig. 3, Ag₂Se droplets are uniformly distributed, showing an average splat diameter of 48 μm. This allows us to delicately optimize the drop spacing of Ag₂Se droplets, which in turn affects the uniformity, porosity, and thickness of the printed patterns. It is found that when the drop spacing is 40 μm, ink consumption is low, but it gives rise to uneven patterns with porosity (in red) as high as 34% (Fig. 1g). Reducing the drop spacing results in uniform print patterns. However, a drop spacing as low as 10 μm deteriorates the resolution of the printed features and imposes higher ink consumption. Consequently, we set the drop spacing at 20 μm in this work, which allows us to obtain

patterns with high resolution. For example, multiple parallel lines with a line width of 150 μm and a gap of 100 μm can be controllably obtained (Fig. 1h). It is worth noting that such high resolution is hardly achievable by commonly used film preparation methods such as spin coating, sputtering, thermal evaporation, and screen printing, reflecting the superiority of inkjet printing technology. Furthermore, we investigated the thickness of the annealed Ag₂Se films in relation to the number of printing layers. Samples with different numbers of printing layers were prepared separately and annealed after printing. As shown in Fig. 1i and Supplementary Fig. 4, the film thickness gradually increases with increasing number of printing layers. However, more printing layers result in a significant discrepancy in the film thickness values. To ensure a uniform film thickness, 40 printing layers are used in subsequent studies.

### Characterization of inkjet-printed Ag₂Se films

XRD patterns (Fig. 2a) of the printed Ag₂Se film annealed at 723 K reveal that the main diffraction peaks can be indexed to β-Ag₂Se phase (PDF#24-1041), indicating an orthorhombic crystal structure. The XRD peaks at 2θ = 23° and 47°, corresponding to the (002) and (004) planes

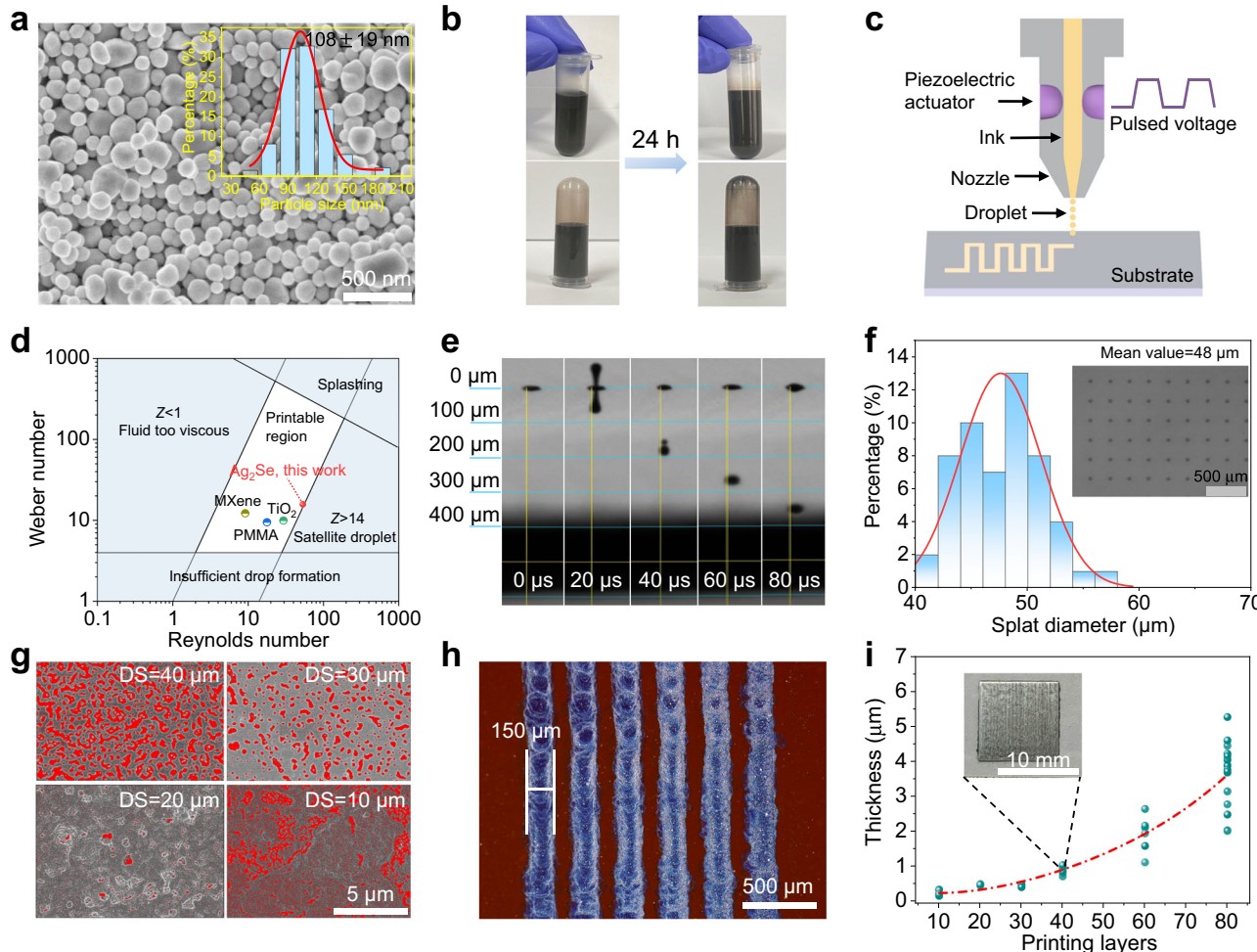

**Fig. 1 | Ag₂Se nanoparticles synthesis and ink printing characteristics.**
**a** Scanning electron microscopy image of Ag₂Se nanoparticles. The inset shows the particle size statistics. **b** Digital photographs of Ag₂Se ink in the normal and inverted states before and after 24 h of resting. **c** Schematic diagram of the working principle of an inkjet printing equipment. An electronically-driven piezoelectric actuator generates a pressure pulse that ejects the droplets from the nozzle.
**d** Parameter plot of the Reynolds and Weber numbers for our Ag₂Se ink. Several inks reported previously for inkjet printing are included for comparison[35,36].

**e** Jetting cycle of a 10 pL Ag₂Se ink droplet during a time interval of 20 μs. **f** Splat diameter histogram for Ag₂Se ink and the droplets on a photographic paper (inset). **g** Dependence of the uniformity and porosity of printed Ag₂Se films on the droplet spacing (DS). **h** Inkjet printing multiple parallel lines on a micron scale using Ag₂Se ink, demonstrating high printing accuracy. **i** Variation of Ag₂Se film thickness with the number of printing layers after annealing. The inset shows a photograph of an Ag₂Se film with 40 printing layers.

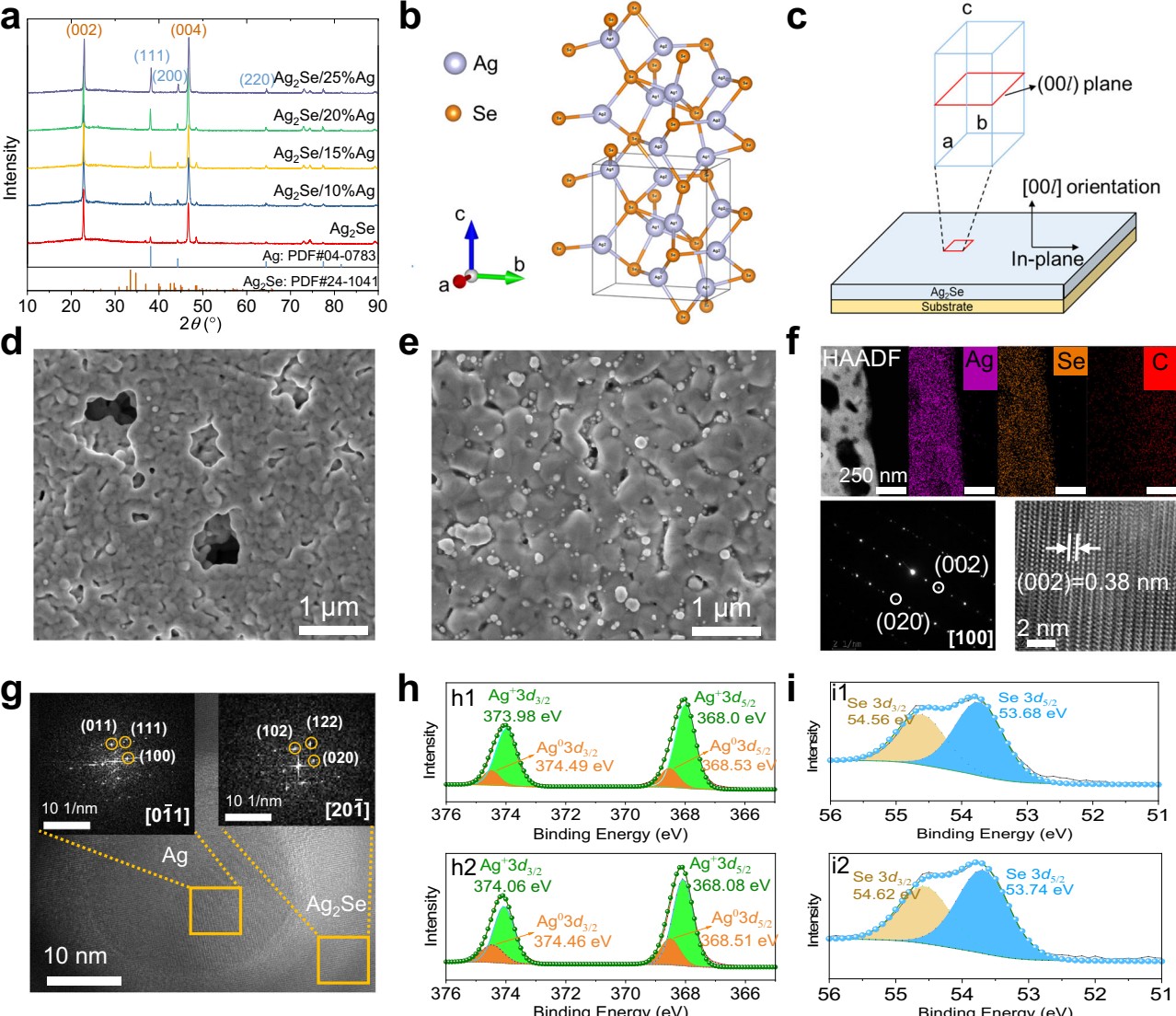

**Fig. 2 | Microstructure and phase composition of inkjet-printed Ag₂Se-based films. a** XRD patterns of Ag₂Se/Ag composite films annealed at 723 K. **b** Crystal structure of low-temperature β-Ag₂Se phase. **c** Schematic diagram of film orientation growth. **d** SEM image of Ag₂Se film. **e** SEM image of Ag₂Se/15%Ag composite film. **f** HAADF-STEM image of the cross-section of Ag₂Se film, and corresponding EDS elemental mapping, together with SAED pattern and HRTEM image. **g** High-resolution TEM image of Ag₂Se/15%Ag composite film with the inset showing FFT images corresponding to Ag₂Se grain and Ag grain. **h** XPS results of Ag 3*d* core-level spectra: h1 for Ag₂Se; h2 for Ag₂Se/15%Ag. **i** XPS results of Se 3*d* core-level spectra: i1 for Ag₂Se; i2 for Ag₂Se/15%Ag.

of β-Ag₂Se, become particularly strong after annealing (Supplementary Fig. 1a). This indicates an increase of crystallinity and the preferential growth of a large number of Ag₂Se grains along the c-axis direction (Fig. 2b, c), similar to the Ag₂Se films prepared by other methods, such as vacuum-assisted filtration[12] and thermal evaporation[29]. Furthermore, the XRD results show the presence of a second phase of Ag in the annealed Ag₂Se film, with diffraction peaks at 2θ = 38° and 44° corresponding to the (111) and (200) planes of cubic Ag (PDF#04-0783). Excess Ag may come from the preceding reduction of Ag ions during solvothermal synthesis[37]. The microstructure and phase composition of the annealed Ag₂Se film were then fully characterized as they determined the thermoelectric properties of the film. SEM images show that the surface of Ag₂Se film is flat, and the distribution of Ag and Se elements is homogeneous (Fig. 2d and Supplementary Fig. 5). However, there are numerous pores with a size of approximately 1 μm inside the film. These pores can seriously deteriorate the electrical properties of the film, as evidenced by its square resistance, which is as high as 14 Ω/□. In order to improve the density of the film and to facilitate the transport of electrons, we then

added commercial inkjet-printable Ag inks with different weight percentages (Supplementary Fig. 6) to the Ag₂Se ink (see Methods). As a result, the addition of Ag ink does not affect the printability of the Ag₂Se ink (Supplementary Table 1), but significantly reduces the porosity of the films after annealing at 723 K (Fig. 2e and Supplementary Fig. 5). The increase in film densities is due, on one hand, to the melting of small-sized Ag nanoparticles during annealing of the film, which acts as a solder to locally connect the Ag₂Se particles[38]; on the other hand, to the direct filling of the pores by those larger-sized Ag particles. XRD patterns (Fig. 2a) indicate that the peak intensity of the Ag phase increases gradually with the addition of commercial Ag ink, but the crystallinity and (00*l*) texture of the Ag₂Se films are not significantly affected.

TEM and scanning TEM (STEM) were conducted to further study the nanostructure characteristics and compositions of the films. A focused ion beam (FIB) was used to prepare cross-sectional TEM samples cut from Ag₂Se and Ag₂Se/15%Ag composite films. The high-angle annular dark field STEM (HAADF-STEM) image and EDS elemental mapping confirm the uniform distributions of Ag and Se in the pristine

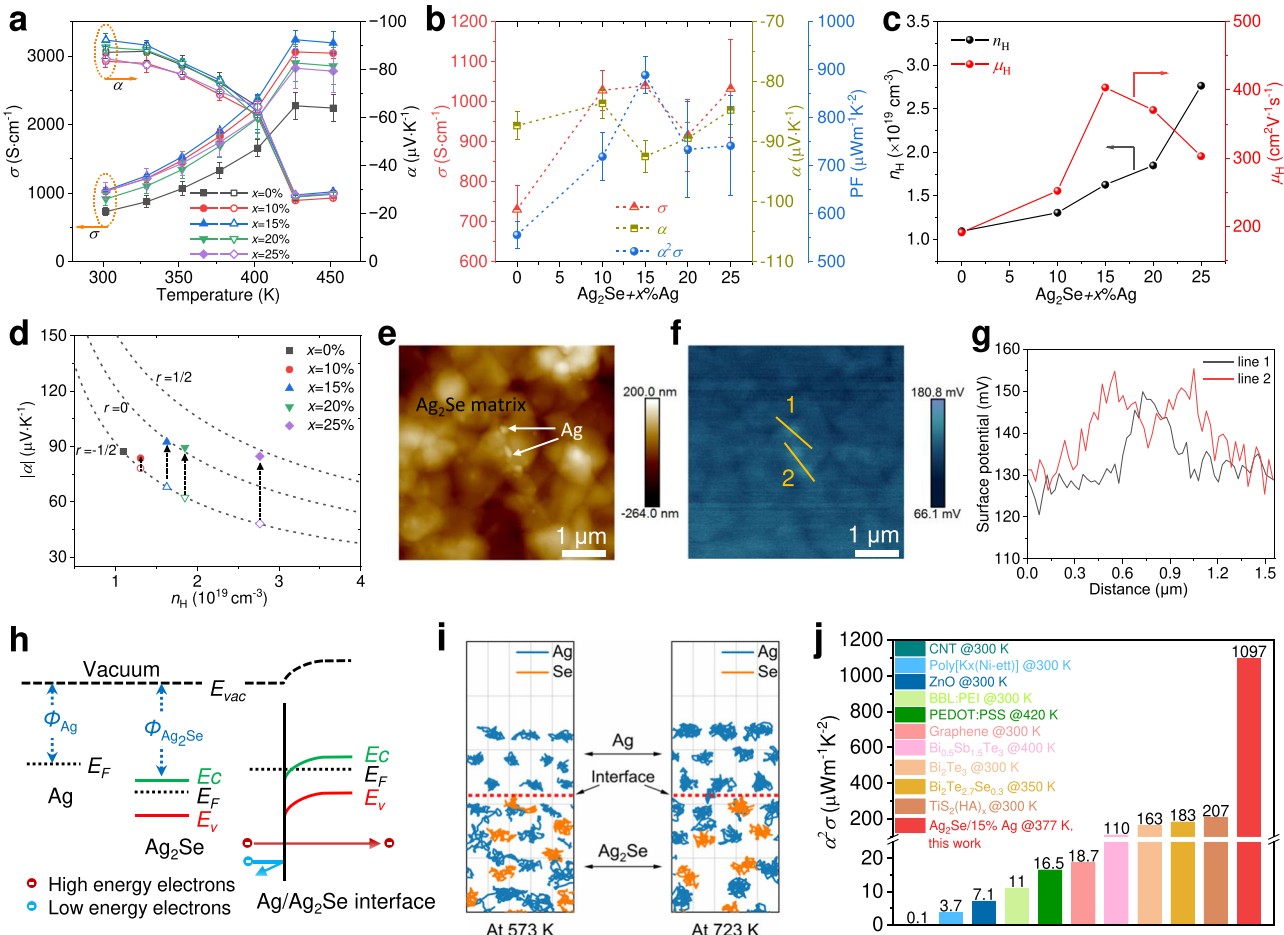

**Fig. 3 | Thermoelectric properties of inkjet-printed Ag₂Se-based films.**
**a** Temperature-dependent electrical conductivity ($\sigma$) and Seebeck coefficient ($\alpha$) ($x$ indicates the weight percentage of Ag ink added). **b** Room-temperature $\sigma$, $\alpha$ and power factor ($\alpha^2\sigma$) as a function of Ag ink content. **c** Room-temperature Hall carrier concentration ($n_H$) and mobility ($\mu_H$). **d** Seebeck coefficient as a function of carrier concentration for Ag₂Se/Ag composite films. **e** AFM topography image of Ag₂Se/ 15%Ag. **f** Surface potential map measured by KPFM. **g** Surface potential line scans of the interfaces between dispersed-Ag particles and Ag₂Se matrix. **h** Energy band diagram for Ag\Ag₂Se before and after contact. **i** Molecular dynamics calculations showing atomic exchanges between Ag and Ag₂Se interface. **j** Comparison of the power factor of inkjet-printed films[11,24–28,40–43].

Ag₂Se film (Fig. 2f). The selected-area electron diffraction (SAED) pattern of Ag₂Se film reveals an orthorhombic lattice viewed along [100] zone axis. High-resolution TEM image proves the good crystallinity and strongly oriented grains along the (00*l*) plane (Fig. 2f), consistent with the XRD patterns. A typical cross-sectional TEM image of an Ag₂Se/15% Ag composite film and the corresponding EDS elemental mapping are shown in Supplementary Fig. 7a. In contrast, the composite film shows a denser microstructure with few pores and more Ag-rich regions, in agreement with SEM observations. EDS line scanning combined with HRTEM confirms the co-existence of orthorhombic Ag₂Se and cubic Ag phases (Fig. 2g and Supplementary Fig. 7b). Also, it can be found that Ag particles are tightly wrapped by Ag₂Se, further confirming that the addition of Ag ink plays a role in filling the voids of Ag₂Se particles during the annealing process. X-ray photoelectron spectroscopy (XPS) results further determine the elemental composition and valence states of the films (Fig. 2h, i, Supplementary Fig. 8 and Tables S2–S4). The spectrum shows that Ag, Se, C, O, and N have been detected in the Ag₂Se and Ag₂Se/15%Ag composite films annealed at 723 K. Notably, the percentage of fitted Ag⁰ spectra increases with the addition of Ag ink, in agreement with XRD results.

### Thermoelectric properties of inkjet-printed Ag₂Se films
The temperature-dependent electrical conductivity ($\sigma$) and Seebeck coefficient ($\alpha$) of inkjet-printed Ag₂Se-based films annealed at 723 K

were simultaneously measured along the in-plane direction from 300 to 460 K. The pristine Ag₂Se film shows an electrical conductivity of 730.7 S·cm⁻¹ and a Seebeck coefficient of −87.3 μV·K⁻¹ (n-type) at room temperature (Fig. 3a). The electrical conductivity increases with increasing temperature while the absolute Seebeck coefficient decreases, and both show a sharp variation around 420 K due to the phase transformation from β-Ag₂Se (semiconductor) to α-Ag₂Se (superionic conductor)[12]. With the addition of Ag ink, the electrical conductivity increases significantly while the Seebeck coefficient is well maintained without obvious deterioration (Fig. 3a). Especially, the room-temperature electrical conductivity increases to 1040.2 S·cm⁻¹ for the Ag₂Se/15%Ag composite film, resulting in a power factor as high as 889.0 μW·m⁻¹·K⁻² (Fig. 3b). Based on the Hall effect measurement, we obtain the carrier concentration ($n_H$) and carrier mobility ($\mu_H$). As shown in Fig. 3c, $n_H$ monotonically increases with increasing Ag ink content, and $\mu_H$ also increases when the loading fraction of Ag ink is not higher than 15 wt%. Therefore, the simultaneous increase in $n_H$ and $\mu_H$ leads to a significant increase in the electrical conductivity. However, an increase in $n_H$ usually leads to a decrease in the Seebeck coefficient (Supplementary Note 2), which is not the case here. To elucidate the reasons for the variation in electrical properties, we plotted the Seebeck coefficient as a function of carrier concentration (Fig. 3d). The dotted lines are Seebeck coefficient as a function of carrier concentration calculated assuming a parabolic

density of states and a simple power-law dependence of the relaxation time (Supplementary Note 3). By assuming acoustic phonon scattering ($r = -1/2$), the increased carrier concentration should result in a much lower Seebeck coefficient for the composites. However, all nanocomposites show an enhanced Seebeck coefficient for a given carrier concentration. This is attributed to the higher scattering parameter in the composites, which compensates for the decrease in the Seebeck coefficient due to the increased carrier concentration. The increase in scattering parameter is usually ascribed to the impurity scattering or barrier energy scattering[39]. Given that the impurity scattering mainly occurs upon doping or filling with alien atoms, which is obviously not the case here, it is very likely that the energy barrier scattering plays a key role for our Ag$_2$Se/Ag samples.

To further analyze the electrical transport properties between the Ag$_2$Se and Ag interfaces, we investigated the local work function ($\Phi$) near the interface by Kelvin probe force microscopy (KPFM). Figure 3e, f shows the topography and the surface potential of Ag$_2$Se/15%Ag sample. According to the surface potential analysis (Fig. 3g), the work function of Ag is ~0.03 eV smaller than that of Ag$_2$Se. Despite the difference in values between the tested and calculated work function (Supplementary Fig. 9), the changing trend keeps the same. The metallic Ag shows a lower $\Phi$ than that of n-type Ag$_2$Se semiconductor, suggesting that electrons can be injected from Ag to Ag$_2$Se, leading to the increase of the carrier concentration. Meanwhile, band bending occurs at the Ag$_2$Se/Ag interface (Fig. 3h), which could scatter the carriers with low energy, leading to increased scattering parameters that restrain the severe reduction of the Seebeck coefficient.

Furthermore, MD calculations (Fig. 3i) show that there are atomic exchanges between Ag and Ag$_2$Se interface at higher temperatures (e.g., 723 K). This implies that strongly bound heterogeneous interfaces could form in the composite films during annealing. We further compared the microstructure and thermoelectric properties of Ag$_2$Se/15%Ag films annealed at different temperatures. As shown in Supplementary Fig. 10, after annealing at 573 K, there is no significant change in the particle size and the film is porous. In contrast, when the printed sample is annealed at 723 K, the porosity is significantly reduced. The increase in film densities is due, on the one hand, to the growth of Ag$_2$Se particles at higher temperatures; on the other hand, to the melting/filling of Ag nanoparticles during annealing of the film. The robust Ag$_2$Se/Ag interfaces combined with the bridging effect of Ag particles as conducting paths facilitate the transport of electrons in the composite films, resulting in enhanced $\mu_H$ at a low loading fraction of Ag ink (Fig. 3c and Table S5). However, the addition of excess Ag ink will strongly scatter the electrons, leading to the decrease of $\mu_H$ (Fig. 3c). As a result, the highest electrical conductivity is obtained in the Ag$_2$Se/15%Ag sample.

As a result of the improved $\sigma$ and the unattenuated $\alpha$, the power factor is substantially improved over the entire measurement temperature range (Supplementary Fig. 11). The highest power factor of 1097 $\mu W m^{-1} K^{-2}$ is obtained at 377 K for the Ag$_2$Se/15%Ag composite film. Compared to other inkjet-printed thermoelectric materials[11,24-28,40-43], our result is remarkably higher (Fig. 3j). Furthermore, we performed 10 consecutive thermal cycling tests on the $\sigma$ and $\alpha$ in the temperature range of 300-373 K to investigate the stability of Ag$_2$Se/15%Ag composite film. The results show that the variations of $\sigma$ and $\alpha$ are less than 6% (Supplementary Fig. 12), indicating that the film is well qualified for application in a variable environment close to room temperature.

## Fully inkjet-printed Ag$_2$Se-based flexible devices

Based on the properties-optimized films, we fabricated fully inkjet-printed flexible thermoelectric devices (Fig. 4a). Silver electrodes are printed firstly, followed by printing thermoelectric legs of Ag$_2$Se/15% Ag in electrically series and thermally parallel. After heat treatment, a flexible device is obtained (See Methods). In contrast to existing processes for patterning flexible thermoelectric legs, such as photolithography[44,45], prefabricated stencil methods[13,30], cutting and pasting processes[12,39], our fully inkjet-printing procedure facilitates the precise integration of thermoelectric legs into patterned electrode arrays, thereby making it possible to fabricate flexible thermoelectric devices in a fast, direct, and cost-effective manner, which is compatible with large-scale manufacturing (Fig. 4b, c). In addition, the shape engineerability of the inkjet printing technology enables the design of complex and versatile thermoelectric legs to optimize heat transfer, as exemplified in Supplementary Fig. 13. More notably, benefiting from the excellent printability of our Ag$_2$Se-based inks, we can precisely control the printing of thermoelectric legs with width and length ranging from millimeters down to micrometers. As shown in Fig. 4b, five flexible Ag$_2$Se-based devices with the same geometrical ratio (the ratio of the leg length to leg cross-sectional area) are printed on the polyimide substrate. The dimensional parameters of these devices are detailed in Supplementary Table S6. Among them, the largest device (Device #1) features a single thermoelectric leg size of 4 mm × 10.5 mm, while the smallest device (Device #5) features a single thermoelectric leg size of 250 μm × 660 μm. The filling factor of our inkjet-printed devices can reach up to 81%, and the density of leg integration gets as high as 125 legs per square centimeter (Fig. 4d). This capability to print high-resolution thermoelectric leg patterns makes it possible to build integrated devices with high packing density, thus enabling high power densities[5].

To demonstrate the power generation performance of the inkjet-printed devices, we measured the output voltage and output power as a function of current at temperature gradient ($\Delta T$) from 10 K to 40 K. Theoretically, the five devices with the same geometrical ratio in Fig. 4b should produce the same open-circuit voltage ($V_{oc}$) and the same maximum output power ($P_{max}$) (Supplementary Note 4). Here, taking Device #1 and Device #2 as examples, the measured $V_{oc}$ and $P_{max}$ are indeed comparable (Fig. 4e and Supplementary Fig. 14). For example, $V_{oc}$ of ~30 mV and $P_{max}$ of ~0.8 μW are obtained for both at a $\Delta T$ of 40 K. In addition, we inkjet-printed devices with different numbers of thermoelectric legs (Supplementary Fig. 15). The test results show that the output voltage and power of the devices increase linearly with the number of thermoelectric legs (Fig. 4f). These results demonstrate a high degree of consistency in material properties and a high degree of controllability in device fabrication for mass production and large-scale applications.

In order to compare the performance of different thermoelectric devices, we calculated the normalized power, which is defined as the maximum output power divided by the area of the device perpendicular to the heat flux and the square of the applied temperature difference[5]. Accordingly, the normalized power of Device #1 and Device #2 is 1.1 μW cm$^{-2}$ K$^{-2}$ and 2.0 μW cm$^{-2}$ K$^{-2}$, respectively. Compared to previously reported in-plane flexible thermoelectric devices, such as various devices fabricated by printing techniques[28,43,46-58], and Ag$_2$Se-based devices fabricated by non-printing processes such as physical vapor deposition processes[29], and vacuum filtration[12,39,59-61], our inkjet-printed flexible Ag$_2$Se-based devices show a significant performance advantage (Fig. 4g). This advantage is attributed to high power factor of our printed materials, the short thermoelectric leg length, as well as high integration density of the devices. Theoretically, Devices #3, #4, and #5 would exhibit higher normalized powers owing to their shorter thermoelectric leg lengths and higher integration densities, but their dimensions are too small to be measured in our laboratory at present.

We then characterized the flexibility and thermal reliability of the fully inkjet-printed devices. Device #2 was subjected to repeat mechanical bending with radii of 4 mm, 3.5 mm, and 3.0 mm (Fig. 4h). 1000 bending cycles were performed for one radius, followed by the next. Reassuringly, the resistance of Device #2 increases by less than 10% after 3,000 cycles (Fig. 4h). The device power-generation performance under different temperatures before and after bending at a bending radius of 4 mm for 1000 times is also compared. As shown in

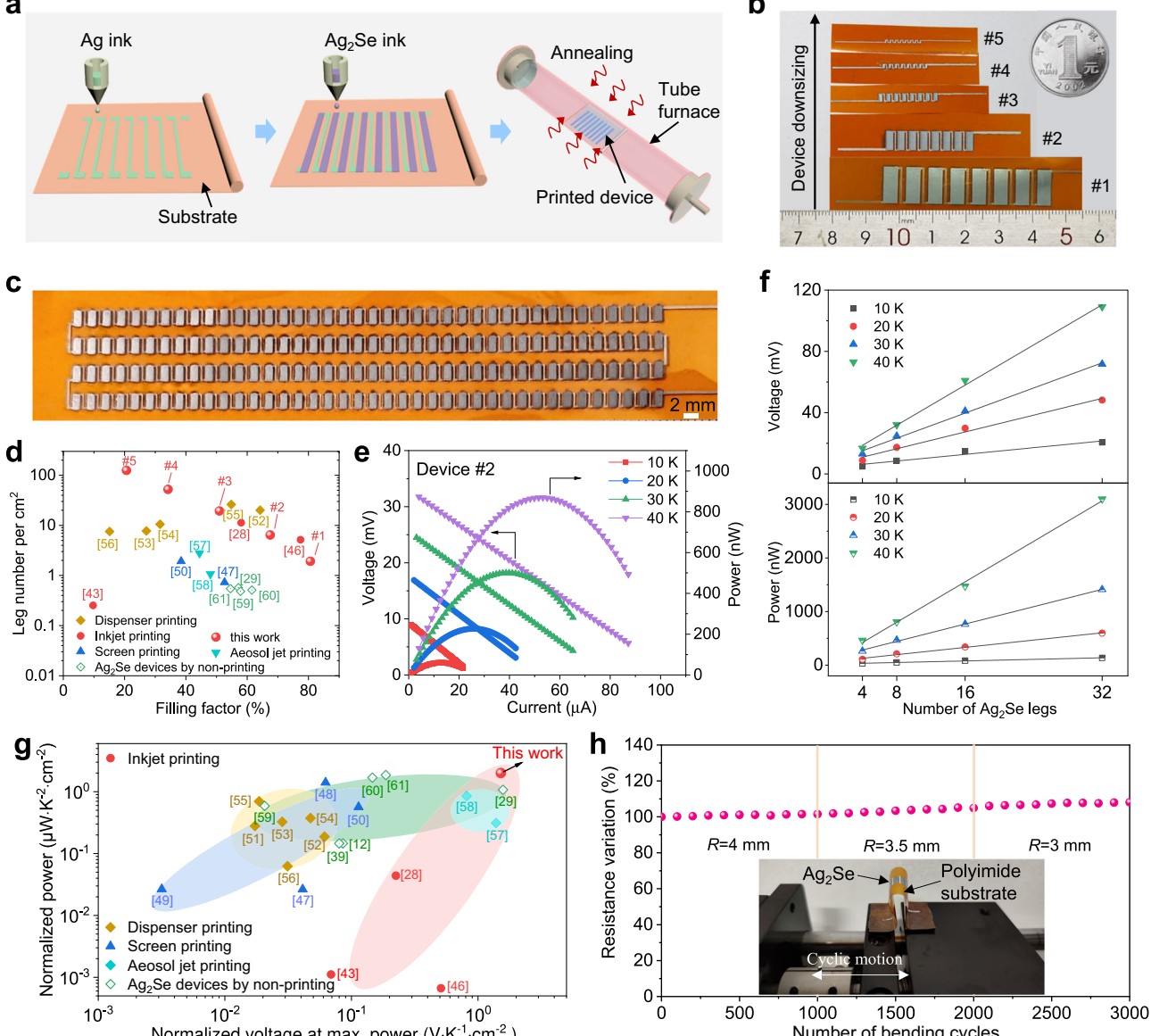

**Fig. 4 | Device fabrication and power generation performance. a** Fabrication flow of a fully inkjet-printed Ag₂Se-based flexible device. **b** Digital photographs of devices consisting of thermoelectric legs with the same geometrical ratio but the width and length ranging from micrometers to millimeters. **c** Digital photograph of a fully inkjet-printed Ag₂Se-based flexible device consisting of 160 Ag₂Se legs. **d** Comparison of the filling factor and leg integration density of state-of-the-art in-plane flexible devices fabricated by different methods. **e** Output voltage and power as a function of current at different temperature gradients. **f** Output voltage and power as a function of the number of Ag₂Se legs. **g** Comparison of the normalized voltage and normalized power of state-of-the-art in-plane flexible devices. **h** Change of device resistance after continuously repeated bending at different bending radii.

Supplementary Fig. 16 and Table S7, the normalized power decays less than 10%, demonstrating superior flexibility. Moreover, the bending test along the direction of the thermoelectric legs was performed, which also shows good flexibility (Supplementary Fig. 17). The excellent flexibility probably results from the (00 *l*)-textured feature of the printed film, as was recently reported in Bi₂Te₃ film[14]. In addition, thermal cycling tests in air were performed on Device #1. The results show a fast response and stable output voltage without degradation over 130 cycles (Supplementary Fig. 18). These test results demonstrate the great potential of our fully inkjet-printed Ag₂Se-based flexible devices for sustainable power generation applications.

**Various power generation scenarios**

Inkjet-printed Ag₂Se-based flexible devices offer the ability to revolutionize the design and fabrication of multi-scale devices while reducing

costs, and opening new applications for the IoT, wearable electronics, and medical devices. As a showcase, we designed and fabricated a solar thermal/thermoelectric/radiative cooling (STR) hybrid device, which integrates a photothermal layer, an inkjet-printed thermoelectric device, and a radiative cooling layer to enable outdoor all-day power generation (Fig. 5a–c). The prototype device includes a ring-like thermoelectric device with fan-shaped thermoelectric legs that are inkjet printed on a flexible polyimide substrate. The outer and inner diameters of the thermoelectric device are 80 and 10 mm, respectively. A carbon-based photothermal material is applied to the outer edge of the thermoelectric device by direct writing. A radiative cooling material cut into a circle is stuck upon the thermoelectric device after a thermal insulation layer is inserted. The properties of the photothermal material and radiative cooling material are shown in Supplementary Fig. 19. This unique structural design makes it possible to set

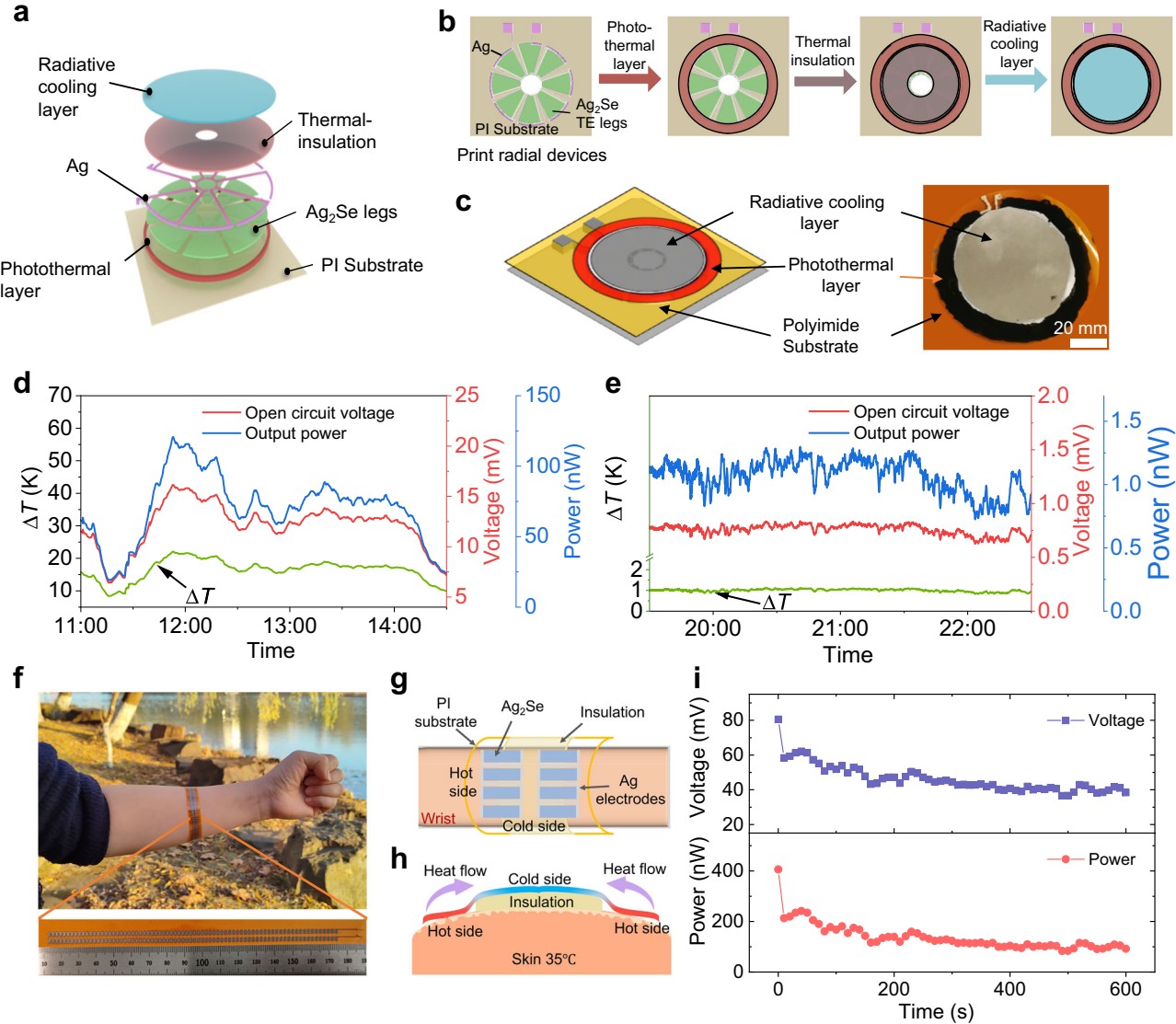

**Fig. 5 | Demonstrations of thermoelectric power generation by harvesting thermal energy from the environment and human body, respectively.**
**a** Exploded schematic diagram of a solar thermal/thermoelectric/radiative cooling (STR) hybrid device. **b** Schematic diagram of the preparation process of a STR hybrid device. **c** Schematic diagram and corresponding optical photograph. Power generation performance of a STR hybrid device: **d** During daytime; **e** At night.
**f** Photograph of generating electricity using the temperature difference between the wrist and the environment. A fully inkjet-printed $Ag_2Se$-based flexible device consisting of 150 $Ag_2Se$ legs is used for the demonstration. Schematic diagram of thermoelectric power generation utilizing body heat: **g** top view; **h** side view. **i** Electricity obtained by using body heat.

the hot and cold ends on the same plane, achieving an effective $\Delta T$ along the in-plane direction (Supplementary Fig. 20). As a result, out-door tests show that a maximum $\Delta T$ of 22 K can be established during the daytime, which yields a corresponding $V_{oc}$ of 16.1 mV and a $P_{max}$ of 120.3 nW (Fig. 5d). Meanwhile, a stable $\Delta T$ of 1 K can be maintained at night, resulting in a $V_{oc}$ of 0.8 mV and $P_{max}$ of 1.3 nW (Fig. 5e). These results prove that this STR hybrid device can achieve 24-hour continuous power generation using the naturally-available temperature differences. The electrical output performance can be further enhanced by integrating more thermoelectric legs and enlarging the device area. Only 8 thermoelectric legs are used here to demonstrate the feasibility. The output voltage and power of STR devices will increase with increasing number of thermoelectric legs, similar to the enhancement shown in Fig. 4f. This hybrid power generation device operates completely passively without any external artificial energy input and requires no maintenance, offering a green and sustainable solution for the energy supply of decentralized wireless sensors and smart IoT devices.

Furthermore, benefiting from the unique patterning capability and high resolution of inkjet printing technology, we printed another device consisting of 150 $Ag_2Se$ legs and then used it to harvest body heat (Fig. 5f–h). As a result, the output power can initially reach 400 nW and then stabilize at 100 nW (Fig. 5i). Correspondingly, the voltage can be maintained above 40 mV. Such power generation performance is capable of running some low-power microelectronics, such as wireless intraocular pressure monitors[62], and 32 kHz Quartz Oscillators[63]. The size of this 150-legged device is only 1 cm in width and 16 cm in length, so 10 of these small devices (similar in size to a tennis wristband) are capable of generating microwatts of power. Considering that the wearable or implantable sensors progressively require lower powers to operate[5], our flexible thermoelectric generators can fully meet the power demand as a portable and sustainable battery, as inkjet printing opens the path to adapting the required voltage and power by adjusting the number of thermoelectric legs.

In summary, we prepared $Ag_2Se$ nanoparticles and formulated them into printable inks. By tuning printing parameters and optimizing

ink formulation, we obtained $Ag_2Se$-based thermoelectric films and flexible devices via inkjet printing technology. The printed $Ag_2Se$/15% Ag composite film exhibited a power factor of 1097 $\mu Wm^{-1}K^{-2}$ at 377 K, more than five times that of reported inkjet-printed materials. In addition, it is found that the additive, non-contact, and maskless nature of inkjet printing provides a simple, inexpensive, and scalable route to pattern complex thermoelectric legs over a large area. Benefiting from the excellent printability of $Ag_2Se$-based inks, we have designed and fabricated a number of flexible devices with leg widths and lengths ranging from millimeters down to micrometers. This unique patterning capability and high-resolution device integration have rarely been reported for in-plane flexible thermoelectric devices, as it is very challenging to achieve via commonly used film preparation methods such as screen printing, dispenser printing, spin coating, sputtering, and thermal evaporation. As a consequence, our devices achieve unprecedentedly high normalized voltage and power. Furthermore, based on inkjet-printed high-performance devices, we demonstrated the applications for different power generation scenarios. Our research strategy can be extended to more thermoelectric materials to synthesize printable inks and fabricate devices with multi-scale dimensions and complex shapes by inkjet printing technology so that flexible thermoelectric generators can be directly integrated into a variety of emerging printed microelectronics (e.g., inkjet-printed displays, inkjet-printed sensors) to serve as power supply units for energy-autonomous systems.

## Methods

### Preparation of inks and inkjet printing

Raw materials used in this work and the synthesis of $Ag_2Se$ nanoparticles are detailed in Supplementary Methods. The synthesized $Ag_2Se$ nanoparticles were then sonicated in 10 mL of ethanol for 2 h to form an $Ag_2Se$ ink at a concentration of 10 mg·mL$^{-1}$. $Ag_2Se$/Ag composite inks were formulated by mixing $Ag_2Se$ inks with a commercial Ag ink (BroadCON INK550, 30 wt%) at different weight percentages. To ensure the homogeneous distribution of Ag particles in the composites, we used ultrasonic dispersion in the form of a solution to formulate $Ag_2Se$/Ag inks. Taking the $Ag_2Se$/15%Ag ink as an example, 25 mg Ag ink was added into 5 mL $Ag_2Se$ ink and then sonicated in a water bath for 2 hours. The composite ink was stored at a low temperature (2–5 °C) to avoid oxidation.

### Inkjet printing of flexible $Ag_2Se$-based films

The prepared ink was then loaded into an inkjet print cartridge of a FUJIFILM Dimatix printing system (DMP-2850). The cartridge has a nozzle diameter of 21 $\mu$m and a nominal droplet volume of 10 pL. The printing pattern was designed by a computer-controlled program. Prior to printing, the nozzle temperature was set to 30 °C. During printing, the substrate platform was heated to 40 °C to accelerate the evaporation of the solvent. The thermoelectric film is built up layer-by-layer. After printing one layer, it is dried for 40 s before printing the next layer. After printing, the sample was first dried under vacuum at 60 °C for 6 h, then placed in a tube furnace and annealed under a mixed gas atmosphere (95%Ar + 5%$H_2$) at 450 °C for 10 minutes. Polyimide films (50 $\mu$m thick) were used as the substrates. Prior to use, the substrates were ultrasonically cleaned with isopropyl alcohol, ethanol, and water for 15 minutes, respectively.

### Inkjet printing of flexible $Ag_2Se$-based devices

Ag electrodes are printed firstly, followed by printing thermoelectric legs of $Ag_2Se$/15%Ag in electrical series and thermally parallel. A droplet spacing of 50 $\mu$m is used for printing Ag electrodes, and a droplet spacing of 20 $\mu$m is used for printing $Ag_2Se$ legs. The printed devices were dried under vacuum at 60 °C for 6 h and then annealed at 450 °C for 10 minutes, which is the same procedure used to print the films above.

### Characterization

Ink viscosity was measured using a rotational viscometer (ROTA-VISC lo-vi Complete) at 25 °C. Surface tension was measured by a contact angle goniometer (Kruss DSA100). X-ray diffraction (XRD) measurements were performed on a Rigaku D/Max-2550 PC (Japan) diffractometer, using Cu-K$\alpha$ radiation ($\lambda = 1.541$ Å) at 40 kV, 30 mA. The microstructure and chemical compositions of $Ag_2Se$ powders and composite films were analyzed by field emission scanning electron microscopy (FE-SEM, TESCAN/MAIA3, Czech), equipped with an energy dispersive spectrometer. Transmission electron microscopy (TEM, JEOLJEM-2100F) was used to further investigate the microscopic morphology of $Ag_2Se$ powder and commercial Ag. The cross-sectional microstructure of the films was examined by a double-aberration corrected transmission electron microscope (Hitachi HF5000 @200 kV in TEM and STEM modes. The TEM samples were prepared by the Focused Ion Beam (FIB, FEI Versa 3D) with the in-situ lift-out technique. X-ray photoelectron spectroscopy measurements were used to investigate the elemental valence states of the films through a Thermo Fisher Scientific electron spectrometer (XPS, ESCALAB 250Xi) with Al K$\alpha$ X-ray beam (1486.6 eV). The spectra fitting was performed using Thermo Avantage software. The high-resolution spectra of the Ag 3d and Se 3d were obtained using C1s as the reference at 284.8 eV. All spectra were fitted with Gaussian peaks after Shirley background subtraction. The optical absorption spectra were evaluated on a UV-Vis-NIR spectrophotometer (HITACHI, U-3010). The surface potential distribution was measured by Kelvin probe force microscopy (Bruker Dimension Icon).

### Performance evaluation

The in-plane electrical conductivity and Seebeck coefficient were simultaneously using a ZEM-3 (ULVAC-RIKO, Japan) at He atmosphere, with a measurement uncertainty of 5%. Hall coefficients ($R_H$) were determined by the Hall measurement system (Lakeshore 8400 Series HMS, USA) using the Van der Pauw method. The power generation performance of the inkjet-printed thermoelectric devices was measured by a homemade apparatus as detailed in our previous report[3]. Repeated bending tests were performed using a mechanical system with a controlled bending radius. Device resistance was measured after every 100 cycles.

### Theoretical calculations

Our calculations were performed on the Vienna Ab initio Simulation Package (VASP). Calculations were conducted based on density functional theory (DFT) within local-density approximations (LDA)[64]. DFT calculations were performed using pseudo-potentials established by the projector-augmented wave (PAW) method and the Perdew-Burke-Ernzerhof (PBE) exchange-correlation functional. Geometry optimization of $Ag_2Se$ and all surface relaxation calculations were performed. The exchange-correlation effects were modeled using the PBE generalized gradient approximation (GGA) functional. All calculations considered the spin effect with the plane wave cutoff energy for 400 eV. The Kohn-Sham equations were solved using a basis of augmented plane wave. Ab initio molecular dynamics (AIMD) was performed in NVT steady-state by Nosé-Hoover thermostat with the 2 fs time steps and 40 ps total simulation time[65]. 10 $Ag_2Se$ with 12 Ag atoms were used to study the atomic exchanges between $Ag_2Se$ and Ag interface at high temperatures.

## Data availability

The authors declare that the data supporting the findings of this study are available within the paper and its supplementary information files and the data that support the findings of this study are available from the corresponding author upon reasonable request.

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

## Acknowledgements
This work was funded by the National Natural Science Foundation of China (No. 51871053, 52174343, U23A20685), the Innovation Program of Shanghai Municipal Education Commission (202101070003E00110), Shanghai Committee of Science and Technology (No. 20JC1415200, 23520710300). A.H. and X.C. acknowledge financial support from the National Natural Science Foundation of China (No. 62175248). H.C. acknowledges financial support from the National Natural Science Foundation of China (No. 52002406). We are grateful to the High-Performance Computing Center of Central South University for partial support of this work. Q.Z. and U.L. acknowledge the Deutsche Forschungsgemeinschaft (DFG, German Research Foundation) under Germany's Excellence Strategy via the Excellence Cluster 3D Matter Made to Order (EXC-2082/1-390761711) for financial support, and acknowledge funding by the European Research Council, grant 101097876 - ORTHOGONAL.

## Author contributions
Q.Z. and L.W. conceived the ideas and designed the work. Y.L. carried out the experiments including material preparation and characterization, device fabrication, and measurements. A.H., Y.W., and S.W. contributed to microstructural characterization. A.H. and K.Z. contributed to the fabrication and characterization of STR hybrid devices. H.Y.C. carried out the density functional theory and molecular dynamics calculations. Y.F. assisted with the power-generation measurements. W.Z. contributed to the drawings. Y.L. and Q.Z. wrote the draft. X.C., L.W., U.L., and W.J. contributed to the discussion and editing. All authors approve the final version of the manuscript.

## Competing interests
The authors declare no competing interests.
