## [Peer Review File · Nature Communications]

Fully inkjet-printed Ag₂Se flexible thermoelectric devices for sustainable power generationREVIEWER COMMENTS

Reviewer #1 (Remarks to the Author):

In this manuscript, the authors have developed Ag₂Se-based thermoelectric films and flexible devices via inkjet printing. Printed Ag₂Se-based films exhibit (001)-textured feature, and an exceptional power factor ($1097 \mu\text{Wm}^{-1} \text{K}^{-2}$ at 377 K) is obtained by engineering the film composition and microstructure. The authors adopted the inkjet printing method to achieve a power factor of $889 \mu\text{Wm}^{-1} \text{K}^{-2}$ at room temperature, which still belongs to the low level in the Ag₂Se-based material. For example, the Ag₂Se films can be prepared by low-cost methods such as screen printing or sputtering with a high power factor (over $1500 \mu\text{Wm}^{-1} \text{K}^{-2}$, DOI: 10.1039/d0ta05859a) at room temperature. In addition, the authors try to improve the thermoelectric properties of the films through the energy filtering and modulating doping mechanisms brought about by silver doping, but the authors' discussion on the energy filtering effect of silver nanoparticles does not agree well with the experimental data. Therefore, it is not suitable for publication in high impact journal.

Some suggestions are as follows:

1. The difference between the work function of silver and silver selenide in Figure 3d is about 0.7eV, and a larger barrier will scatter more electrons, and Seebeck will increase significantly and the conductivity will decrease. It is inconsistent with the experimental results of Seebeck coefficient decline when silver is doped by 10% in Fig. 3 b, and the increasing conductivity is also contrary to the energy filtering effect, thus these contradictions need to be reasonably explained. In addition, in Fig. 3c, nH monotonically increases with increasing Ag ink content, and μH also increases when the loading fraction of Ag ink is not higher than 15wt%. Here, the reasons for the increased μH when Ag ink is less than 15wt% should be explained.
2. In figure 5, proposed thermal/thermoelectric/radiative cooling device (STR) under the condition of temperature difference of 40 K can only output of about 100 nW is extremely weak power, far short of drivers need dozens of μW power consumption level of microelectronic device. If worn on a human wrist whose temperature difference with the outside world is only about 3K, the output power is almost negligible, so the device prepared in this paper does not have practical value.
3. Solar thermal/thermoelectric/radiative cooling (STR) hybrid device should be given real product photo displayed on supporting materials.

Reviewer #2 (Remarks to the Author):

The authors show the Ag₂Se-based flexible thermoelectric devices interestingly fabricated by the inkjet-printing technique. Such a technique enables a controllable fabrication of the devices with a variation in dimension from millimeters to micrometers. Moreover, the composition optimization using 15%Ag-inclusion leads to a high power factor in Ag₂Se-based composite films, and achieves a high normalized output power density. This work offers a facile method for the fabrication of the flexible thermoelectric devices, and many audiences would be interested in this. The manuscript is well organized and the experimental results well support the discussion and conclusion, while there are still many concerns should be addressed.

- (1) The increase in Ag concentration leads to a simultaneous increase in carrier concentration and carrier mobility as the concentration lower than 15%, but the carrier mobility decreases as further increasing Ag concentration. What is the mechanism for this result? How to ensure the homogeneous distribution of Ag particles in the films.
- (2) It is known that Bi₂Te₃-based thermoelectrics show the highest performance near room temperature, which have been fabricated by the 3D printing technique. What are the advantages for the inkjet printing technique?
- (3) Ag₂Se-based flexible thermoelectric devices fabricated by different methods have been widely investigated, which should be referred for the comparison.
- (4) For the bending test, the bending direction should be perpendicular to that of the current test.

Dear Reviewers,

Thank you very much for taking the time to review our manuscript NCOMMS-23-32636 entitled "Fully inkjet-printed Ag₂Se flexible thermoelectric devices for sustainable power generation". We have carefully revised the manuscript according to your constructive comments and suggestions. For ease of reference, your comments and suggestions are reproduced in **blue** and our responses are in **black** in this response letter. Our changes to the manuscript are highlighted in **red** in the revised manuscript and revised supporting information.

Please see our point-by-point responses below.

REVIEWER COMMENTS

Reviewer #1 (Remarks to the Author):

In this manuscript, the authors have developed Ag₂Se-based thermoelectric films and flexible devices via inkjet printing. Printed Ag₂Se-based films exhibit (001)-textured feature, and an exceptional power factor ($1097 \mu\text{Wm}^{-1}\text{K}^{-2}$ at 377 K) is obtained by engineering the film composition and microstructure. The authors adopted the inkjet printing method to achieve a power factor of $889 \mu\text{Wm}^{-1}\text{K}^{-2}$ at room temperature, which still belongs to the low level in the Ag₂Se-based material. For example, the Ag₂Se films can be prepared by low-cost methods such as screen printing or sputtering with a high power factor (over $1500 \mu\text{Wm}^{-1}\text{K}^{-2}$, DOI: 10.1039/d0ta05859a) at room temperature. In addition, the authors try to improve the thermoelectric properties of the films through the energy filtering and modulating doping mechanisms brought about by silver doping, but the authors' discussion on the energy filtering effect of silver nanoparticles does not agree well with the experimental data. Therefore, it is not suitable for publication in high impact journal.

Response:

We highly appreciate your professional comments. With regard to your first question related to material properties, we totally agree with you that some Ag₂Se films with higher power factors have been reported. However, **the performance of existing Ag₂Se-based devices prepared using these films is still poor.** This is mainly

attributed to two reasons. One is that those films prepared by methods such as sputtering, thermal evaporation or vacuum filtration **exhibit poor control of the thickness and roughness and have limited design flexibility and low levels of device integration.** On the other hand, **the films are prepared independently of the device.** As a result, performance is greatly reduced when the films are assembled into the devices.

To solve these problems, we thereby focus on inkjet printing. Using inkjet printing, we can easily realise a variety of patterned designs and prepare small-sized, highly integrated miniature devices. For example, we printed a number of flexible devices with leg widths and lengths ranging from millimeters down to micrometers (**Fig. R1a**). The filling factor of our inkjet-printed devices can reach up to 81%, and the density of leg integration gets as high as 125 legs per square centimeter (**Fig. R1b**). **This unique patterning capability and high-resolution device integration has hardly been reported for flexible thermoelectric devices because it is extremely challenging to achieve this through commonly-used film preparation methods such as spin coating, sputtering, thermal evaporation, and screen printing.** As a consequence, our devices achieve unprecedentedly high normalized voltage and power (**Fig. R1c**).

Figure R1| a, Digital photographs of devices consisting of thermoelectric legs with the same geometrical ratio but the width and length ranging from micrometers to millimeters. **b**, Comparison of the filling factor and leg integration density of state-of-the-art in-plane flexible devices fabricated by different methods. **c**, Comparison of the normalized voltage and normalized power of state-of-the-art in-plane flexible devices.

In addition, we are able to inkjet print devices with different numbers of thermoelectric legs (**Fig. R2a**). The test results show that the output voltage and power of the devices increase linearly with the number of thermoelectric legs (**Fig. R2b** and **2c**). We have also prepared a device with 160 Ag_2Se legs (**Fig. R3**). The low-cost printing technique and excellent device performance demonstrate the ability of mass production and the potential for large-scale applications.

Figure R2| a, Digital photograph of fully inkjet-printed Ag_2Se -based flexible device consisting of different number of Ag_2Se legs. **b**, Output voltage as a function of the number of Ag_2Se legs. **c**, Output power as a function of the number of Ag_2Se legs.

Figure R3| Digital photograph of a fully inkjet-printed Ag₂Se-based flexible device consisting of 160 Ag₂Se legs.

Moreover, we are very grateful to you for pointing out the shortcomings of our discussion on the energy filtering effect. In this regard, we have further analyzed the relationship between Seebeck coefficient and carrier concentration at different scattering factors. In addition, we have also investigated the local work function near the interface of Ag₂Se and Ag by Kelvin probe force microscopy. Please see the answer to your question #1 below for detailed discussion.

Some suggestions are as follows:

#1. The difference between the work function of silver and silver selenide in Figure 3d is about 0.7 eV, and a larger barrier will scatter more electrons, and Seebeck will increase significantly and the conductivity will decrease. It is inconsistent with the experimental results of Seebeck coefficient decline when silver is doped by 10% in Fig. 3b, and the increasing conductivity is also contrary to the energy filtering effect, thus these contradictions need to be reasonably explained. In addition, in Fig. 3c, n_H monotonically increases with increasing Ag ink content, and μ_H also increases when the loading fraction of Ag ink is not higher than 15wt%. Here, the reasons for the increased in μ_H when Ag ink is less than 15wt% should be explained.

Response:

Thank you very much for your insightful questions. The work function obtained from the density functional theory (DFT) calculation is too idealized, so there will be some discrepancy between the calculated value and the real value. In order to accurately analyze the electrical transport properties between the Ag₂Se and Ag interfaces, we investigated the local work function near the interface by Kelvin probe force microscopy (KPFM). **Fig. R4a** and **b** shows the topography and the surface potential of Ag₂Se/15%Ag sample. According to the surface potential analysis (**Fig. R4c**), the work function of Ag is ~0.03 eV smaller than that of Ag₂Se. Despite the difference in values between the tested and calculated work function, the changing trend keeps the

same. In this regard, the electron charge can transfer from Ag to Ag₂Se, leading to the increase of the carrier concentration. Meanwhile, band bending occurs at the Ag₂Se/Ag interface, which could scatter the carriers with low energy, leading to increased scattering parameter that restrains the severe reduction of Seebeck coefficient.

Figure R4 | **a**, AFM topography image of Ag₂Se/15%Ag. **b**, Surface potential map measured by KPFM. **c**, Surface potential line scans of the interfaces between dispersed-Ag particles and Ag₂Se matrix.

The evidence of low-energy electron scattering in our nanocomposites is shown by plotting the Seebeck coefficient as a function of carrier concentration in **Fig. R5**. The dotted lines are Seebeck coefficient as a function of carrier concentration calculated assuming a parabolic density of states and a simple power-law dependence of the relaxation time:

$$\alpha = \pm \frac{k_B}{e} \left[\frac{\left(r + \frac{5}{2}\right) F_{r+3/2}(\zeta^*)}{\left(r + \frac{3}{2}\right) F_{r+1/2}(\zeta^*)} - \zeta^* \right] \quad (1)$$

$$n = \frac{N_V^{2/3}}{2\pi^2} \left(\frac{2k_B T m_d}{\hbar^2} \right)^{3/2} F_{1/2}(\zeta^*) \quad (2)$$

where k_B is Boltzmann's constant, e is the electronic charge, $\zeta^* = E_F / (k_B T)$ is the reduced Fermi energy, m_d is the density of states effective mass, N_V is the number of degenerate valleys, r is the energy-dependent relaxation time exponent and F_s is the Fermi integral given by

$$F_s(\zeta^*) = \int_0^\infty x^s [\exp(x - \zeta^*) + 1]^{-1} dx \quad (3)$$

As is shown in **Fig. R5**, by assuming acoustic phonon scattering ($r = -1/2$), the increased carrier concentration should result in much lower Seebeck coefficient for the composites. However, all nanocomposites show an enhanced Seebeck coefficient for a

given carrier concentration. This is attributed to the higher scattering parameter in the composites, which compensates for the decrease in Seebeck coefficient due to the increased carrier concentration. Therefore, the Seebeck coefficient is well maintained without obvious deterioration after the addition of Ag ink. The increase in scattering parameter is usually ascribed to the impurity scattering or barrier energy scattering in principle (Yang, *et al. Npj Comput. Mater.* 2016, 2, 15015; Lu *et al., Adv. Energy Mater.* 2019, 1902986). Given that the impurity scattering mainly occurs upon doping or filling with alien atoms, which is obviously not the case here, it is very likely that the energy barrier scattering plays a key role for our Ag₂Se/Ag samples.

Figure R5 | Seebeck coefficient as a function of carrier concentration for Ag₂Se/Ag composite films.

Furthermore, we can see that n_H monotonically increases with increasing Ag ink content, as electrons can be injected from Ag to Ag₂Se. In contrast, μ_H increases firstly and then decreases with the increasing loading fraction of Ag ink. The enhancement of μ_H at low loading fraction of Ag is ascribed to the bridging effect of Ag particles as conducting paths that facilitate the transport of electrons. However, addition of excess Ag ink will strongly scatter the electrons, leading to the decrease of μ_H . As a result, the highest electrical conductivity is obtained in the Ag₂Se/15%Ag sample.

Thank you again for pointing out these critical issues. We have added the new data and corresponding discussion to the revised manuscript.

#2. In figure 5, proposed thermal/thermoelectric/radiative cooling device (STR) under the condition of temperature difference of 40 K can only output of about 100 nW is extremely weak power, far short of drivers need dozens of μW power consumption level of microelectronic device. If worn on a human wrist whose temperature difference with the outside world is only about 3K, the output power is almost negligible, so the device prepared in this paper does not have practical value.

Response:

Thank you very much for raising concerns about the practical use of our inkjet-printed devices. With regard to the STR, we used an inkjet-printed device consisting of only 8 thermoelectric legs here to demonstrate its feasibility, so the power obtained is only about 100 nW. The output power of STR will increase with increasing number of thermoelectric legs, similar to the enhancement shown in **Fig. R2** above. Therefore, a significant improvement in output power will be achieved if more thermoelectric legs are used.

To verify this, we inkjet-printed a new device consisting of 150 Ag_2Se legs and then used it to harvest body heat (**Fig. R6**). As a result, the output power can initially reach 400 nW and then stabilize at 100 nW. Correspondingly, the voltage can be maintained above 40 mV. Such power generation performance is capable of running some low-power microelectronics, such as wireless intraocular pressure monitors (*Hasan et al. Adv. Mater. Technol. 2021, 6, 2000771*), and 32 kHz Quartz Oscillators (*Fagas et al, ICT-Energy Concepts for Energy Efficiency and Sustainability, 2017*). The size of this 150-legged device is only 1 cm in width and 16 cm in length, so 10 of these small devices (similar in size to a tennis wristband) are capable of generating microwatts of power.

In addition, our inkjet-printed devices here are made only of n-type Ag_2Se . We believe that if both n-type and p-type thermoelectric materials are inkjet printed in the future, substantially higher power will be realized.

We are very grateful to you for pointing out this forward-looking question. We have added these new data and the discussions to revised manuscript.

Figure R6| a, Photograph of generating electricity using the temperature difference between the wrist and the environment. Schematic diagram of thermoelectric power generation utilizing body heat: **b**, top view; **c**, side view. **d**, Electricity obtained by using body heat.

3. Solar thermal/thermoelectric/radiative cooling (STR) hybrid device should be given real product photo displayed on supporting materials.

Response:

Thank you very much for your constructive suggestion. We have supplemented the schematic diagram and corresponding optical photograph (**Fig. R7**). They have been added to Figure 5 of the revised manuscript.

Figure R7| a, Exploded schematic diagram of a novel solar thermal/thermoelectric/radiative cooling (STR) hybrid device. **b**, Schematic diagram of the preparation process of a STR hybrid device. **c**, Schematic diagram and corresponding optical photograph.

All in all, we greatly appreciate your insightful comments on our paper and for your constructive suggestions to improve our work. These remarks have helped us to refine this manuscript, and have also provided us with valuable guidance for our future research. Thank you very much for your selfless contributions.

Reviewer #2 (Remarks to the Author):

The authors show the Ag₂Se-based flexible thermoelectric devices interestingly fabricated by the inkjet-printing technique. Such a technique enables a controllable fabrication of the devices with a variation in dimension from millimeters to micrometers. Moreover, the composition optimization using 15%Ag-inclusion leads to a high power factor in Ag₂Se-based composite films, and achieves a high normalized output power density. This work offers a facile method for the fabrication of the flexible thermoelectric devices, and many audiences would be interested in this. The manuscript is well organized and the experimental results well support the discussion and conclusion, while there are still many concerns should be addressed.

Response:

Thank you very much for recognizing the significance of our work. Your comments are very valuable and helpful to us in revising and improving our paper. We have made revisions in response to each comment.

(1) The increase in Ag concentration leads to a simultaneous increase in carrier concentration and carrier mobility as the concentration lower than 15%, but the carrier mobility decreases as further increasing Ag concentration. What is the mechanism for this result? How to ensure the homogeneous distribution of Ag particles in the films.

Response:

Thank you very much for pointing this out. The change of electrical conductivity due to the addition of Ag ink results from the change of carrier concentration (n_H) and carrier mobility (μ_H). n_H monotonically increases with increasing Ag ink content, as electrons can be injected from Ag to Ag₂Se. In contrast, μ_H increases firstly and then decreases with the increasing loading fraction of Ag ink. The enhancement of μ_H at low loading fraction of Ag is ascribed to the bridging effect of Ag particles as conducting paths that facilitate the transport of electrons. However, addition of excess Ag particles will strongly scatter the electrons, leading to the decrease of μ_H . As a result, the highest electrical conductivity is obtained by the Ag₂Se/15%Ag sample.

Moreover, to ensure the homogeneous distribution of Ag particles in the composites,

we used ultrasonic dispersion in the form of a solution to formulate $\text{Ag}_2\text{Se}/\text{Ag}$ inks. First, the as-prepared Ag_2Se powder was ultrasonically dispersed in ethanol to form a dispersion solution, and then different weight percentages of commercial Ag ink were mixed with the Ag_2Se ink. Then, the mixed ink was ultrasonically treated in a water bath for 2 hours. In order to verify the uniformity of Ag particles, we performed SEM tests on the $\text{Ag}_2\text{Se}/\text{Ag}$ composites. As shown in **Fig. R8**, Ag nanoparticles are uniformly dispersed in the Ag_2Se matrix without obvious agglomeration.

Figure R8| SEM and corresponding EDS mapping of annealed $\text{Ag}_2\text{Se}/15\%\text{Ag}$ film.

(2) It is known that Bi_2Te_3 -based thermoelectrics show the highest performance near room temperature, which have been fabricated by the 3D printing technique. What are the advantages for the inkjet printing technique?

Response:

Thank you very much for your insightful question. Inkjet printing is particularly advantageous in the manufacture of miniature thin-film devices with high packing density. For example, we can easily print a number of flexible Ag_2Se devices with leg widths and lengths ranging from millimeters down to micrometers (**Fig. R1a, above**). The filling factor of our inkjet-printed devices can reach up to 81%, and the density of leg integration gets as high as 125 legs per square centimeter (**Fig. R1b, above**). This unique patterning capability and high-resolution device integration has hardly been reported for flexible thermoelectric devices because it is extremely challenging to achieve this through commonly-used film preparation methods such as screen printing, dispenser printing, spin coating, sputtering, and thermal evaporation. As a consequence, our devices achieve unprecedentedly high normalized voltage and power (**Fig. R1c, above**).

Our research strategy can be extended to more thermoelectric materials to synthesize printable inks and fabricate devices with multi-scale dimensions and complex shapes by inkjet printing technology, so that flexible thermoelectric generators can be directly integrated into a variety of emerging printed microelectronics (e.g. inkjet-printed displays, inkjet-printed sensors) to serve as power supply units for energy-autonomous systems.

Thank you very much for raising this issue. We have included this in the conclusion section of the revised manuscript.

(3) Ag₂Se-based flexible thermoelectric devices fabricated by different methods have been widely investigated, which should be referred for the comparison.

Response:

Thank you very much for your constructive suggestion. We have added the performance of Ag₂Se-based devices prepared by non-printing methods for comparison. Please refer to **Figure R1** above. The corresponding figure in the main text of revised manuscript has been replaced. Thanks a lot for your suggestion.

(4) For the bending test, the bending direction should be perpendicular to that of the current test.

Response:

Thank you very much for pointing this out. We have supplemented the bending experiments of printed devices in the other direction. As shown in **Fig. R9a-c**, the device resistance increases by 36% after 1000 bending cycles at a bending radius of 3.5 mm.

In terms of wearable requirements, the TE device is worn on the human arm as a wristband. Therefore, the bending direction of our f-TEG is usually perpendicular to the direction of the thermoelectric legs (**Fig. R9d**). Given this consideration, we have kept the original bending test results in the main text and added this new bending test to the supplementary information.

Figure R9| a, Change of the device resistance after continuously repeated bending at a bending radius of 3.5 mm. **b**, and **c**, Photographs of bending tests. **d**, Photograph of generating electricity using the temperature difference between the wrist and the environment. A fully inkjet-printed Ag_2Se -based flexible device consisting of 150 Ag_2Se legs is used for the demonstration.

In all, we are very grateful to you for evaluating our paper and for your constructive comments to improve our work. Your dedication is much appreciated.

REVIEWERS' COMMENTS

Reviewer #1 (Remarks to the Author):

I think the authors have made mainly satisfactory responses and revisions to improve the paper, and my main concerns have been met and the paper looks publishable.

Reviewer #2 (Remarks to the Author):

The authors have well answered the comments and provided additional experiment results to support the discussions and conclusions. So I recommend to publish the manuscript in this version.

REVIEWER COMMENTS

Reviewer #1 (Remarks to the Author):

I think the authors have made mainly satisfactory responses and revisions to improve the paper, and my main concerns have been met and the paper looks publishable

Response:

Thank you very much for recognizing our work and for your insightful comments to improve our paper. Your dedication is much appreciated.

Reviewer #2 (Remarks to the Author):

The authors have well answered the comments and provided additional experiment results to support the discussions and conclusions. So I recommend to publish the manuscript in this version.

Response:

Thank you very much for your positive comments and your recommendation.